# Assessment and Spatial Modelling of Agrochernozem Properties for Reclamation Measurements

**Ruslan Suleymanov** [1,2], **Azamat Suleymanov** [3], **Gleb Zaitsev** [1,2], **Ilgiza Adelmurzina** [4,5], **Gulnaz Galiakhmetova** [4,5], **Evgeny Abakumov** [6,*] and **Ruslan Shagaliev** [5]

1    Ufa Institute of Biology, Ufa Federal Research Centre, Russian Academy of Sciences, 69, October pr-t., 450054 Ufa, Russia
2    Laboratory of Climate Change Monitoring and Carbon Ecosystems Balance, Ufa State Petroleum Technological University, 1, Kosmonavtov st., 450064 Ufa, Russia
3    Department of Environmental Protection and Prudent Exploitation of Natural Resources, Ufa State Petroleum Technological University, 1, Kosmonavtov st., 450064 Ufa, Russia
4    Department of Geodesy, Cartography and Geographic Information Systems, Ufa University of Science and Technology, 32, Zaki Validi st., 450076 Ufa, Russia; gulnazgizatshina@yandex.ru (G.G.)
5    Laboratory of Geoinformation Systems in the Field of Decarbonization, Ufa State Petroleum Technological University, 1, Kosmonavtov st., 450064 Ufa, Russia
6    Department of Applied Ecology, Saint-Petersburg State University, 16-Line V.O. 29, 199178 Saint-Petersburg, Russia
*    Correspondence: e_abakumov@mail.ru or e.abakumov@bio.spbu.ru

**Abstract:** Traditional land-use systems can be modified under the conditions of climate change. Higher air temperatures and loss of productive soil moisture lead to reduced crop yields. Irrigation is a possible solution to these problems. However, intense irrigation may have contributed to land degradation. This research assessed the ameliorative potential of soil and produced large-scale digital maps of soil properties for arable plot planning for the construction and operation of irrigation systems. Our research was carried out in the southern forest–steppe zone (Southern Ural, Russia). The soil cover of the site is represented by agrochernozem soils (Luvic Chernozem). We examined the morphological, physicochemical and agrochemical properties of the soil, as well as its heavy metal contents. The random forest (RF) non-linear approach was used to estimate the spatial distribution of the properties and produce maps. We found that soils were characterized by high organic carbon content (SOC) and neutral acidity and were well supplied with nitrogen and potassium concentrations. The agrochernozem was characterized by favorable water–physical properties and showed good values for water infiltration and moisture categories. The contents of heavy metals (lead, cadmium, mercury, cobalt, zinc and copper) did not exceed permissible levels. The soil quality rating interpretation confirms that these soils have high potential fertility and are convenient for irrigation activities. The spatial distribution of soil properties according to the generated maps were not homogeneous. The results showed that remote sensing covariates were the most critical variables in explaining soil properties variability. Our findings may be useful for developing reclamation strategies for similar soils that can restore soil health and improve crop productivity.

**Keywords:** agricultural land; reclamation potential; basic soil features; soil organic carbon; soil quality rating

## 1. Introduction

Research on the Earth's climate change indicates a warming trend [1,2]. Predictive modelling of future temperature change shows further climate warming due to intensive human-made impacts [3]. Climate change is transforming native ecosystems [4], including agricultural systems, which are exceptionally important in the context of the increasing population of the Earth to provide sufficient food of good quality [5].

As the climate warms, the frequency, duration and intensity of droughts are expected to increase. Such conditions will significantly limit crop production options and reduce crop yields [6,7]. For example, rice yields in China [8] and Korea [9], cotton and sorghum in South Africa [10] and olives in Portugal [11] are threatened.

This raises the need for climate-smart agriculture [12]. Irrigation is one such effective method [13]. Soil irrigation is essential for ensuring the healthy growth of crops and plants. Irrigation helps to supply water to the root zone of plants, which is critical for their survival and growth. Moreover, complementary irrigation is critical to ensuring crop yields and agricultural quality in many regions of the Earth [14,15]. Under these conditions, irrigation is ranked first among other aspects of soil formation, as its intensification leads to changes in the physical [16], water [17], physicochemical [18,19], chemical [20,21] and biological [22,23] soil regimes. Thus, irrigation is an influential factor in transforming and transferring matter and energy in a soil profile.

The main requirements for reclamation and irrigation measures are assessing soil conditions and creating large-scale up-to-date maps. The latter requirement is necessary to study and understand the spatial distribution of soil nutrients. The use of machine learning techniques in digital soil mapping is now being actively implemented [24]. One of the most popular methods is the random forest (RF) classification algorithm, which consists of many decision trees [25]. This algorithm has demonstrated reliable results for digital mapping of soil properties at different scales [26–28]. For example, Wiesmeier et al. [29] applied the RF approach for the spatial prediction of soil organic carbon (SOC) stocks in Inner Mongolia, Northern China. The authors showed that land use, soils and geology were the most important variables influencing SOC storage. Similarly, this method was applied to digital mapping of SOC and calcium carbonate equivalent contents, where it was found that the key factors for the distribution of properties were remote sensing data [30].

In the context of climate change, proper soil irrigation is even more critical as it can help to mitigate the impacts of drought and extreme weather events, which are becoming more frequent and severe due to climate change. Efficient use of water through soil irrigation can help to conserve water resources, reduce greenhouse gas emissions associated with water pumping and transportation and enhance soil carbon sequestration, which can help to mitigate climate change. Climate change equally occurs in forest–steppe zones, including in the study region [31,32]. Such conditions make it necessary to increase the share of irrigated land and design new irrigation systems. Hence, detailed land cover studies and mapping are needed in planning irrigation system construction and operations to avoid degradation processes (water and wind erosion, accumulation of toxic salts, erosion of soil structure and loss of SOC and nutrients). The study's primary purpose is to assess the soils for use in irrigated agriculture in the forest–steppe zone of the Republic of Bashkortostan (Russia). Therefore, the objectives of this study are the following: (i) study the physicochemical properties of soils, (ii) assess the geochemical condition of soils, and (iii) conduct the digital mapping of soil properties using the RF approach in combination with remote sensing and topography data.

## 2. Materials and Methods

### 2.1. Site Description

The study site (327 hectares) is an arable plot located in the southern forest–steppe zone, on the Belaya River's second floodplain terrace (Figure 1). The study area is characterized by flat terrain and is located near the river, which made it suitable for involvement in arable farming and further irrigation. The soil cover of the site is represented predominantly by agrochernozem soils (Luvic Chernozem). There are also small areas of meadow–swamp soils. Meadow–swamp soils are formed in drainless declines and karst craters. When the meltwater and the soil–ground water close to the surface are stagnant, with gleying processes developing. These soils are not plowed because they are unsuitable for agricultural use. For these reasons, no more specific study of meadow–swamp soils has been conducted.

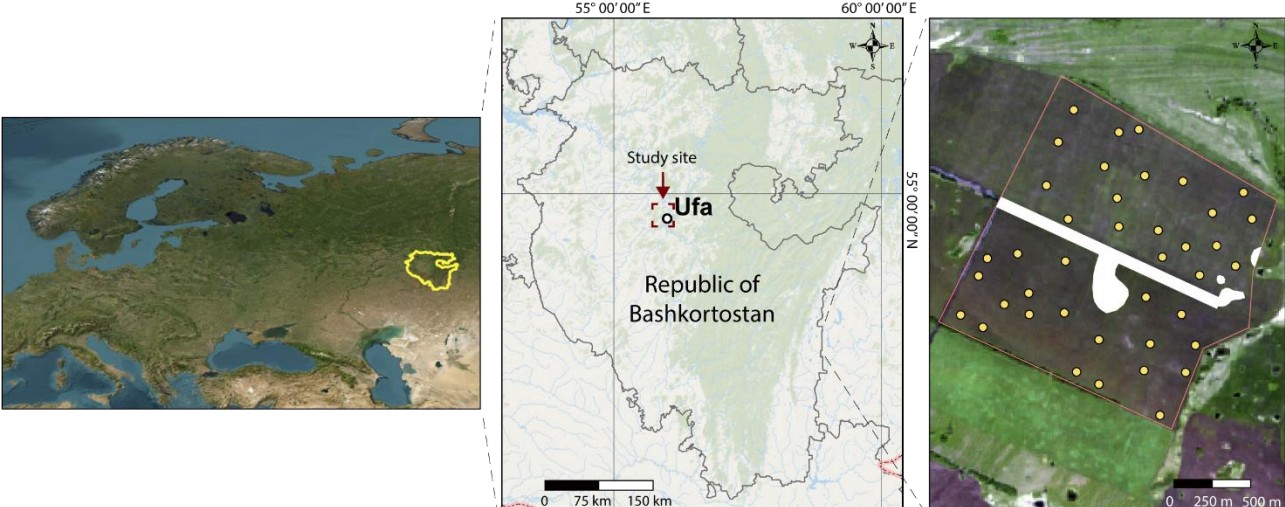

**Figure 1.** Location of the study site and distribution of samples (image source: Sentinel-2A—natural color composite).

The climate of the study area is described as continental and moderately arid. The average annual air temperature is 2.8 °C, the mean January temperature is −15 °C and the absolute minimum is −46 °C. The mean July temperature is +19 °C, and the absolute maximum is +38 °C. Droughts are frequent in spring, autumn and the second half of the summer.

The terrain of the study area is generally levelled; the gradients are small, with a slope predominance of 1–2° (0–1° = 112 ha (34.2%); 1–2° = 172 ha (52.6%); 2–4° = 42 ha (12.8%); 4–6° = 1 ha (0.4%)), but there are karst funnels. The height above sea level varies from 80 to 90 m. Groundwater is found at a depth of 8 m; water composition is hydrocarbon–calcium–magnesium with a 0.3 g/L. Water for irrigation would be taken from the Belaya River. The water composition is hydrocarbon–calcium with mineralization of 0.3–0.6 g/L.

### 2.2. Data Collection and Chemical Analysis

For this study, we excavated 15 full-length sections on the site. Soil samples were collected from each established genetic horizon over the entire width of the soil profile [33]. Additionally, 40 soils were sampled from the upper horizon (0–10 cm) to assess the spatial distribution of soil properties. The selected samples were air-dried, ground and carefully passed through a 1 mm sieve. Soil descriptions in the field and further agrochemical analyses were carried out using conventional methods [34–36]. Under laboratory conditions, SOC content, soil acidity (pH), soluble salts, alkaline hydrolyzed nitrogen (N), available phosphorus ($P_2O_5$) and potassium ($K_2O$), exchangeable calcium ($Ca^{2+}$) and magnesium ($Mg^{2+}$) cations and exchangeable sodium ($Na^+$) were determined.

The water–physical properties of soils were measured according to Shein and Karpachevskii guidelines [37]. The heavy metal content was determined in conformity with the methodological guidelines [38]. Soil availability criteria for organic matter, nutrients and micronutrients were assessed by Kiryushin [39]. The index of soil quality and yield was defined by Karmanov et al. [40].

Soil categorization and the characteristics of the ground profile and layers were accomplished considering substantive genetic classification [41]. The World Reference Base duplicated the consequent soil type name for Soil Resources [42].

### 2.3. Random Forest

We applied the RF algorithm for the spatial prediction of soil properties because it is the most popular in digital soil mapping studies [43]. Moreover, this method is superior to other machine learning methods, as shown in a variety of studies [28,44–47]. RF is an

ensemble learning method for classification and regression tasks. In a model, each decision tree in the forest is built using a random subset of the input data and a random subset of the input features. Then, the results of all individual trees are aggregated to make a single prediction. We defined the default parameters required by the RF model: the number of trees to be built in the forest (ntree) and the number of predictors to be used in each tree-building process (mtry). We set the parameters as follows: ntree = 500 and mtry = 4.

The recursive feature elimination (RFE) method was applied to select an optimal set of environmental auxiliary variables among all covariates [48]. The algorithm works by backward selection, where the least promising predictors are excluded from the model based on an initial predictor importance measure.

### 2.4. Environmental Variables

The environmental covariates used for digital mapping included Sentinel-2A satellite data (S2A) and topographic attributes. S2A data were selected during the period with minimal vegetation (bare soil). The dataset of 25 May 2019, which was used for study, matched these parameters. Satellite bands and the Normalized Difference Vegetation Index (NDVI) were used. Relief attributes were calculated using a 30 m resolution digital relief model (NASA's Shuttle Radar Topography Mission (SRTM), https://www2.jpl.nasa.gov/srtm (accessed on 20 January 2023)). The topographical attributes included elevation, aspect, slope, multiresolution of ridge top flatness (MrRTF) and (MrVBF) multiresolution valley bottom flatness indices. The environmental variables used for modelling are presented in Table 1.

**Table 1.** Environmental covariates used for digital mapping.

| Environmental Covariates | Acronym | Spatial Resolution (m) | Definition | Central Wavelength (nm) |
|---|---|---|---|---|
| Sentinel-2A bands and indices | B2 | 10 | Blue | 492.4 |
| | B3 | 10 | Green | 559.8 |
| | B4 | 10 | Red | 664.6 |
| | B5 | 20 | Red edge 1 | 704.1 |
| | B6 | 20 | Red edge 2 | 740.5 |
| | B7 | 20 | Red edge 3 | 782.8 |
| | B8 | 10 | NIR 1 | 832.8 |
| | B8a | 20 | NIR 2 | 864.7 |
| | B11 | 20 | SWIR 1 | 1613.7 |
| | B12 | 20 | SWIR 2 | 2202.4 |
| | NDVI | 10 | Normalized Difference Vegetation Index | - |
| Topographic attributes (SRTM) | El | 30 | Elevation (m) | - |
| | Aspect | 30 | Aspect (%) | - |
| | Slope | 30 | Slope angle (%) | - |
| | MrRTF | 30 | Multiresolution of ridge top flatness index | - |
| | MrVBF | 30 | Multiresolution Valley Bottom Flatness index | - |

### 2.5. Validation and Statistical Analyses

A leave-one-out cross-validation (LOOCV) approach was applied to evaluate the prediction performance of RF models. LOOCV cross-validation is appropriate for a small dataset and consists of using all training data leaving one out. The accuracy of the predictive model was estimated using a mean error (ME) and a root-mean-square error (RMSE) (Equations (1) and (2)):

$$ME = \frac{\sum_{i=0}^{n}(O_i - P_i)^2}{n} \tag{1}$$

$$RMSE = \sqrt{\frac{\sum_{i=0}^{n}(O_i - P_i)^2}{n}} \qquad (2)$$

where $O_i$ and $P_i$ are observed and predicted values of soil properties, respectively, and $n$ is the number of samples.

Statistical data processing, digital mapping and model accuracy assessment were performed in R and RStudio.

## 3. Results

The agrochernozem soils under study formed on well-watered sections on carbonate eluvial–diluvial clays and heavy loams. The morphological structure characteristics of those soil profiles were that the $A_1$ humus-accumulative horizon was an average of $\approx$84 cm, and the medium depth of a layer $A_1$ + ABca holds $\approx$125 cm. Horizon A was divided into the upper plowable part (Aplow) with a thickness of $\approx$28 cm and the plough-pan part ($A_1$). We described the soil section on the flat area (Table 2).

**Table 2.** Description of soil horizon.

| Horizon (Depth, cm) | Description of Soil Horizon |
|---|---|
| Aplow 0–28 | Dark grey, almost black, moist, powdery lumpy, heavy loamy, medium dense, many roots, transition through ploughing line is unnoticeable. |
| $A_1$ 28–84 | Dark grey, moist, grainy, heavy loamy, medium compacted, gradual transition. |
| $A_1$Bca 84–125 | Dark brown with grey shade, wet, coarse, heavy loamy, mycelium carbonate, effervescence from 10% HCl, transition is noticeable. |
| Bca 125–150 | Brown, unevenly colored with humus shades in the directions of roots, moist, lumpy, heavy loamy, impermeable, carbonates in the form of mycelium and softened large grains, effervescence from 10% HCl, transition is gradual. |
| Cca 150–180 | Yellowish brown, wet, unsolid, large-cloddy, heavy clay, mycelium carbonates and softened large white grains, effervescence from 10% HCl. |

The morphological description of the soil profile showed that the humus-accumulative horizon (A) was characterized by a dark grey or almost black color. The texture of the Aplow, $A_1$ and $A_1$Bca horizons was classified as heavy loamy. There were many roots at the toplayer. The top plowable part's structure was powdery–lumpy, the $A_1$ plough-pan part was granular and the $A_1$Bca horizon became coarse-grained. The upper bound for detecting carbonates as a mycelium occurred inside the $A_1$Bca horizon. The Bca horizon was characterized by a lumpy structure that becamelarge-cloddy within the Cca layer.

The analysis of soil chemical properties is presented in Table 3. The soils were characterized by a neutral reaction in the upper horizons (pH $H_2O$ 6.6–6.9), but with depth, the reaction varies to weakly alkaline (pH $H_2O$ 8.3). The average SOC content was 4.4% in the plowable horizon and decreased by 3.7% in A1 and 0.6% in the C layer. The content of hydrolyzable nitrogen ranged from 186 to 26 mg/kg (from Aplow to Cca layer) and exchangeable potassium from 152 to 54 mg/kg, which, in the studied soils, was classified as "high" according to grading [41]. However, the values of available phosphorus were insignificant (51–17 mg/kg). The values of $Ca^{2+}$ (34.7–27.3 mg/kg) and $Mg^{2+}$ (7.7–5.4 mg/kg) varied approximately in the same ranges across all horizons. The base saturation values ranged from 98% in the top layer to 100% in the Cca horizon, while the range of solids values ranged from 0.05 to 0.02% and also tended to decrease with depth.

**Table 3.** Description of soil horizon [1].

| Horizon (Depth, cm) | pH (H₂O) | C org, % | N Alkaline Hydrolyzable, mg/kg | Exchangeable Cations | | Available | | Base Saturation, % | Solids, % |
|---|---|---|---|---|---|---|---|---|---|
| | | | | Ca²⁺ | Mg²⁺ | P₂O₅ | K₂O | | |
| | | | | cmol(+)/kg | | mg/kg | | | |
| Aplow 0–28 | 6.6 ± 0.4 | 4.4 ± 0.7 | 186 ± 29 | 34.7 ± 3.5 | 7.7 ± 1.3 | 51 ± 14 | 152 ± 24 | 98 | 0.05 ± 0.01 |
| A₁ 28–84 | 6.7 ± 0.4 | 3.7 ± 0.7 | 154 ± 39 | 32.3 ± 3.0 | 7.1 ± 0.4 | 53 ± 16 | 157 ± 25 | 94 | 0.04 ± 0.03 |
| A₁Bca 84–125 | 6.9 ± 0.6 | 2.5 ± 0.3 | 114 ± 28 | 29.2 ± 3.9 | 6.2 ± 0.9 | 43 ± 17 | 129 ± 18 | 94 | 0.04 ± 0.01 |
| Bca 125–150 | 7.3 ± 0.8 | 0.9 ± 0.3 | 52 ± 17 | 27.3 ± 5.6 | 6.2 ± 0.9 | 46 ± 12 | 127 ± 18 | 96 | 0.04 ± 0.01 |
| Cca 150–180 | 8.3 ± 0.1 | 0.6 ± 0.2 | 26 ± 4 | 29.6 ± 1.4 | 5.4 ± 0.7 | 17 ± 2 | 54 ± 8 | 100 | 0.02 ± 0.01 |

[1] ±standard deviation, *n* = 15.

Conducive aquatic and physical properties are crucial indicators of the suitability of soils for irrigated agriculture [49,50]. These properties influence soil water, air and nutrient status, crop yields and agricultural product quality [51,52]. The bulk density of the humus-accumulative horizon of the investigated soils was optimal for most crops (1.02 g/cm³ in the to player and 1.44 g/cm³ in Cca) (Table 4). Simultaneously, with depth, the density gradually increases, an illuvial horizon was compacted (1.39 g/cm³), and a soil-forming rock was dense (1.44 g/cm³).

**Table 4.** Water–physical properties of the agrochernozem soils [1].

| Horizon (Depth, cm) | Bulk Density, g/cm³ | Hardness, kg/cm² | Porosity | Wilting Point, % | Field Moisture Capacity | Capillary Moisture Capacity | Total Moisture Capacity | Sand (1–0.05 mm) | Silt (0.05–0.001 mm) | Clay (<0.001 mm) |
|---|---|---|---|---|---|---|---|---|---|---|
| | | | | | | | % | | | |
| Aplow 0–28 | 1.02 ± 0.02 | 1.32 ± 0.02 | 58.9 ± 2.2 | 14.2 ± 0.2 | 43.0 ± 3.7 | 45.0 ± 4.7 | 53.7 ± 1.9 | 12.7 ± 0.1 | 65.9 ± 0.5 | 21.4 ± 0.4 |
| A₁ 28–84 | 1.04 ± 0.03 | 3.13 ± 0.07 | 58.1 ± 2.7 | 14.3 ± 0.1 | 44.8 ± 1.5 | 45.7 ± 1.3 | 52.1 ± 1.3 | 7.8 ± 0.2 | 68.8 ± 4.3 | 23.4 ± 2.1 |
| A₁Bca 84–125 | 1.22 ± 0.001 | 6.89 ± 0.23 | 44.4 ± 0.2 | 13.5 ± 0.4 | 35.1 ± 1.2 | 36.1 ± 0.5 | 40.8 ± 0.9 | 13.6 ± 2.9 | 58.5 ± 6.8 | 27.9 ± 3.8 |
| Bca 125–150 | 1.39 ± 0.007 | 14.99 ± 1.14 | 35.1 ± 0.08 | 12.2 ± 0.2 | 26.0 ± 0.1 | 27.3 ± 1.6 | 30.1 ± 0.9 | 14.9 ± 0.8 | 53.9 ± 0.7 | 31.2 ± 0.1 |
| Cca 150–180 | 1.44 ± 0.03 | 19.99 ± 1.61 | 32.8 ± 1.6 | 11.8 ± 0.1 | 23.9 ± 1.8 | 25.9 ± 3.3 | 29.3 ± 1.0 | 14.9 ± 1.1 | 56.2 ± 0.9 | 28.9 ± 0.1 |

[1] ±standard deviation, *n* = 15.

The hardness of the soil lumps also increased with depth. With the optimal soil texture, humus-accumulative horizons possess a significant rate of air space porosity. The water capacity in the plowable layer was equal to 43.0% (field moisture capacity), 45.0% (capillary moisture capacity) and 53.7% (total moisture capacity). With depth, these values decreased to 23.9, 25.9 and 29.3% in the Cca horizon, respectively. The wilting point of the A₁ humus-accumulative horizon was 14.2–14.3% and decreased to 11.8% with depth. The content of silt (56.2–65.9%) was dominant in the composition of all soil horizons; then, clay and sand contents were descending (21.4–28.9 and 7.8–14.9, respectively).

Figure 2 shows water infiltration of the agrochernozem. Determining water infiltration showed that the average speed of water absorption from the soil surface was 1.52 mm/min and from a depth of 50 cm—2.41 mm/min, which, according to the classification of Kachinsky [37], belongs to the medium and high categories of water infiltration, respectively.

Land irrigation increases the likelihood of contamination of soils and agricultural products by various pollutants, including heavy metals [53,54]. Toxicants may also become more mobile in the soil profile [55]. The analysis of micronutrients and heavy metals showed low and medium cobalt, zinc and copper contents (Table 5). The average values of the heavy metals and micronutrient contents were 2.71 mg/kg for Pb, 0.18 mg/kg for Cd, 0.024 mg/kg for Hg, 0.13 mg/kg for Co, 0.33 mg/kg for Zn, 0.11 mg/kg for Cu and 11.27 mg/kg for Mn at the Aplow layer. The Pb, Cd and Hg contents were well below the maximum allowable concentrations. It can be assumed that if there were no

pollutants in the water used for irrigation, there will be no risk of contamination from agricultural produce.

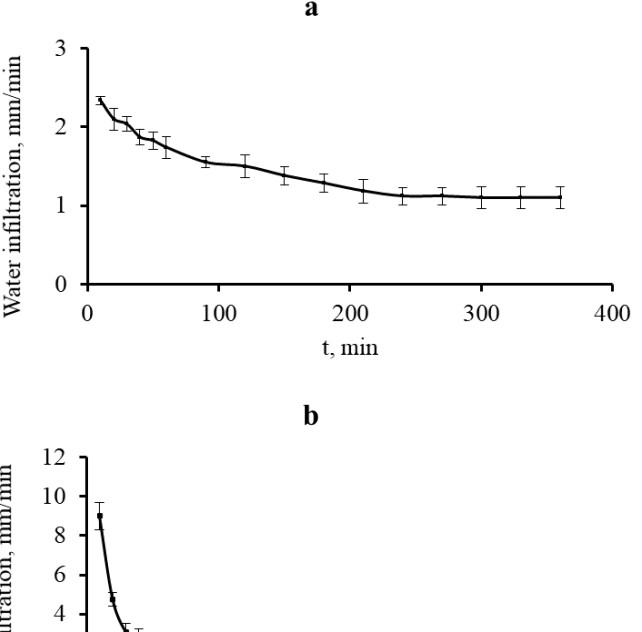

**Figure 2.** Water infiltration of the agrochernozem studied: (**a**) water absorption speed from the soil surface; (**b**) water absorption speed from a depth of 50 cm.

**Table 5.** Heavy metals and micronutrient content [1].

| Horizon (Depth, cm) | Pb | Cd | Hg | Co | Zn | Cu | Mn |
|---|---|---|---|---|---|---|---|
| | mg/kg | | | | | | |
| Aplow, 0–28 cm | 2.71 ± 0.36 | 0.18 ± 0.02 | 0.024 ± 0.003 | 0.13 ± 0.01 | 0.33 ± 0.05 | 0.11 ± 0.01 | 11.27 ± 1.11 |
| Level of detection | <M.A.C. [2] (6.0 mg/kg) | <M.A.C. (2.0 mg/kg) | <M.A.C. (2.1 mg/kg) | low content | low content | low content | medium content |

[1] ±standard deviation, $n = 15$; [2] M.A.C.—maximum allowable concentration.

The effectiveness of soil element prediction models using the RF approach according to the error metrics (ME and RMSE) is presented in Table 6. As an indicator of site fertility, the SOC prediction model showed the RMSE value of 0.60, comparable to that of other authors [56,57]. The RMSE values for pH, Mg, Ca, N, P and K were 0.48, 1.56, 2.51, 2.92, 1.57 and 4.57, respectively.

**Table 6.** Test performance of the RF for soil properties at 0–10 cm depth.

| Soil Parameter | ME | RMSE | Top 3 Important Variables |
|---|---|---|---|
| SOC | 0.36% | 0.60% | B12, NDVI, Aspect |
| pH | 0.23 | 0.48 | Slope, Aspect, Elevation |
| Mg | 2.43 cmol(+)/kg | 1.56 cmol(+)/kg | Slope, MRVBF, B2 |
| Ca | 1.58 cmol(+)/kg | 2.51 cmol(+)/kg | B11, B4, MRVBF |
| N | 8.53 mg/kg | 2.92 mg/kg | NDVI, B2, Aspect |
| P | 2.47 mg/kg | 1.57 mg/kg | B12, B8A, Elevation |
| K | 20.88 mg/kg | 4.57 mg/kg | B8A, Aspect, B11 |

The top three important variables for each soil feature were different. The most important variables to explain the spatial distribution of SOC, Ca, N, P and K contents were remote sensing data, while the terrain covariates were responsible for pH and Mg prediction. For example, S2A B12 band, NDVI and aspect were responsible for the SOC variation. Since there was a close positive correlation between the SOC content and N (r = 0.86), the key predictors of these parameters were practically the same. Terrain attributes were less responsible for agrochemical property variations, which can be explained by the relatively flat topography of the plots, low pedologic diversity and insufficient spatial resolution of the DEM. The variation in pH levels was entirely controlled by the slope, aspect and elevation covariates. The slope variable was also the leading predictor of Mg content. The spatial distribution of the remaining elements was controlled by both types of variables.

The generated digital maps of soil properties in the upper horizon (0–10 cm) using the RF method are shown in Figure 3. The spatial distribution of the elements was not homogeneous. The elevated concentrations of pH, Ca, P and K were found mainly in the northern part, while Mg content was found in the southern part of the plot.

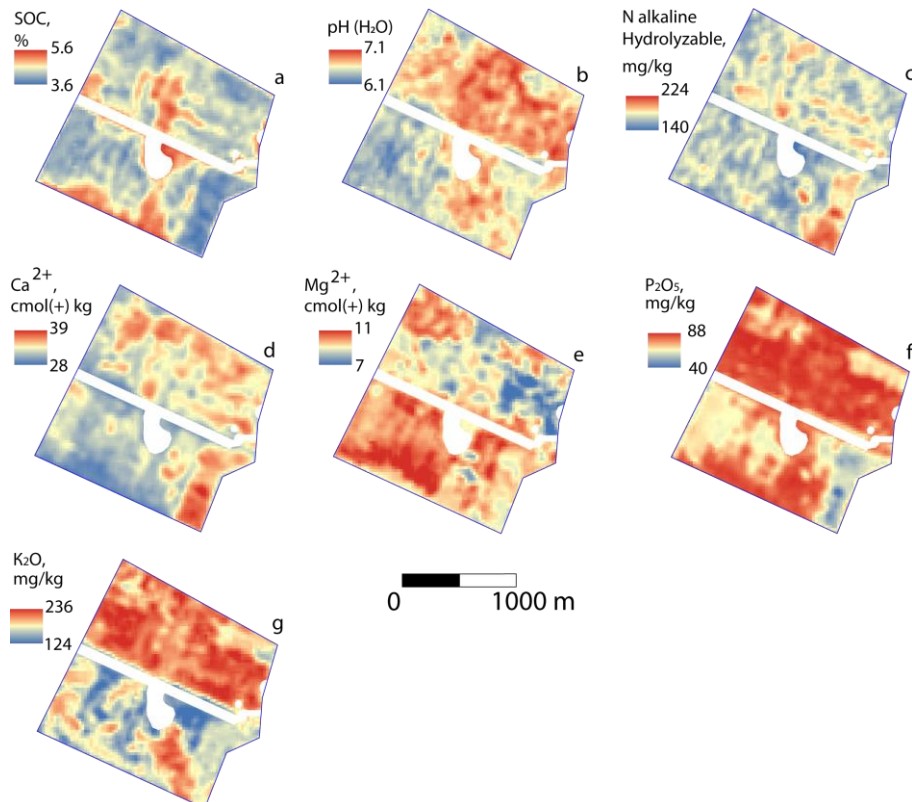

**Figure 3.** The spatial distribution of soil agrochemical properties using the random forest method. Areas with vegetation are masked by white color. SOC (**a**); pH (**b**); N (**c**); Ca (**d**); Mg (**e**); P (**f**); and K (**g**).

## 4. Discussion

The assessment and digital mapping of soil properties for reclamation measurements is a complex process that requires careful planning and implementation. By using a combination of soil analysis, spatial mapping, modelling and monitoring, it is possible to identify the factors that contribute to soil degradation and develop effective reclamation strategies to improve soil health and productivity.

The chemical and water–physical properties of the studied agrochernozem soils correspond to analogues in this region. For example, the granulometric composition was also classified as heavy loam in horizons Aplow, $A_1$, AB and B, while the wilting point values ranged from 12.7% in the plough horizon to 10.6% in the lower one [58]. The pH and SOC values were also comparable with our results. However, the concentrations of $P_2O_5$ and

$K_2O$ in the arable horizon were higher, which is probably due to the application of fertilizers. According to Figure 2, in the upper layer, the rate of water absorption was gradual due to heavy loamy texture and high SOC content. The lower horizons (>50 cm) were clayey, compacted, without structural aggregates and accordingly flow poorly. A sharp spike in water absorption in the lower horizon is associated with the fractured structure of the soil profile and karst. However, the absorption of water stops due to the clay structure. Therefore, it is necessary to set watering rates for irrigation.

Long-term irrigation risk salinization of soils and alteration of cation composition [55,58], resulting in increased toxicity and structural breakdown. In our study, soils were saturated with calcium-dominated bases. Exchangeable sodium in absorbed cations was not detected. This soil is highly resistant to anthropogenic effects [39]. For example, earlier, it was shown that despite the deterioration of the properties of arable chernozem as a result of prolonged conventional plowing, they remain more resistant to degradation compared to other soils [59]. The water-soluble salt content in the profile is relatively low, and the soil is characterized as not saline. It is likely that hydrocarbon–calcium water and irrigation regimes do not pose a threat to soil salinization. However, the irrigation process and soil condition should be monitored. Previously, the effect of long-term irrigation on the properties of Luvic Chernozem in this region has been shown [60]. It was found that agrophysical, chemical and physicochemical properties of the soil deteriorated, and salt content increased over a 30-year period of irrigation.

The content of heavy metals did not exceed the maximum permissible concentrations, which was comparable with other studies in this region. For instance, average values of Pb, Cd and Zn were 5.1, 0.28 and 1.48, respectively, in the arable layer of chernozem, although the Mn values were half as much [61].

As an integral indicator of ground properties [62], a soil quality rating based on criteria such as humus-accumulative capacity, acidity, SOC content and mobile phosphorus showed that the soil quality index was 74 points, with a maximum of 100 points for the reference soil (Calcic Chernozem). Given the crop, the soil quality index was 62 for wheat production and 65 for sunflower grain production. Here, the decrease in the quality index was due to a deficit of productive soil moisture.

Lamichhane et al. [46] after analyzing 120 studies concluded that vegetation indices were found to be more influential in predicting SOC in small plots. At the same time, we hypothesize that soil spectral reflectance explains a key influence on the SOC prediction, as darker color corresponds to increased organic carbon, and therefore N, compared to paler color [63]. Thus, numerous studies have shown good results for spatial prediction of soil properties on arable land using remote sensing data [64,65]. However, Suleymanov et al. [66] applied a Support Vector Machine method in combination with terrain attributes for modeling soil properties on arable chernozem. The authors achieved RMSE coefficients of 5.33 for Ca, 0.73 for P and 32.34 for K. In another study on the digital mapping of agrochernozems properties, the authors reached RMSE values of 4.4, 27.5 and 14.6 for nitrogen, phosphorus and potassium, respectively, using machine learning methods in combination with remote sensing data and collocated soil parameters maps [67]. Thus, Tziachris et al. [68] previously showed that the measured collocated soil parameters that were used as covariates (clay, electric conductivity and pH) had more influence for predicting soil organic matter than the environmental parameters due to flat terrain. In another research, it was reported that the key variables for explaining the spatial patterns of SOC concentration were a clay map, remote sensing data and distance from the river [69]. Similarly, Were et al. [70] showed that a total nitrogen map was the most important variable for the spatial assessment of SOC stocks. Thus, different maps of various soil parameters can be useful information for spatial prediction at different depths in future studies.

External influences on soils directly affect qualitative and quantitative indicators, as well as their spatial variability. Since irrigation directly affects soil cover and its properties, we assume that the spatial distribution of properties will also be subject to change. For example, Biswas and Mojid [71] showed that SOC, total nitrogen and phosphorus

concentrations increased significantly as a result of wastewater irrigation in the topsoil layer. The study conducted on three forage farms irrigated for more than 30 years in Saudi Arabia showed that most soil properties were characterized by moderate or strong spatial dependence [72]. At the same time, changes in soil variability can also be accompanied by fertilization, erosion, plot divisions, crop rotations and change or alternation of tillage practices [73,74]. Thus, permanent irrigation can affect the content and spatial variability of soil parameters, which requires more frequent digital mapping.

Further work to improve the predictive accuracy can be achieved by applying additional variables to explain the variation in properties, unmanned aerial vehicles (UAV) [75,76], predictive techniques that take into account spatial dependence [77] and collecting more soil samples. For example, remote sensing data, such as S2A, has a spatial resolution of 10 m, which may not be sufficient for the digital mapping of soil properties in small areas. Specifically, Baltensweiler et al. [27] concluded that microtopography has the most significant impact on soil acidity variability. Moreover, UAV provides more flexibility in the frequency of the survey and minimizes the effects of weather conditions such as cloud cover [78]. Nevertheless, the main limitation of UAV is the spatial coverage, which makes it difficult to map soil properties at the regional level. Nevertheless, the application of UAV seems to be a more rational solution for assessing the spatial distribution of soil properties on irrigated lands.

## 5. Conclusions

Global climate change is occurring on all parts of the Earth and is accompanied by an increase in the frequency of natural disasters. The trend of increasing average annual temperatures and decreasing precipitation creates a dangerous and unstable situation for agriculture. In this regard, irrigation will be expanded and actively applied, which will certainly affect the qualitative and quantitative characteristics of soils. Therefore, such fertile soils and the dynamics of their change under the influence of irrigation should be examined and studied.

In this study, we examined the soil cover of the arable plot designated for irrigation in the southern forest–steppe zone of Russia (Southern Ural). The evaluation of morphological, physicochemical and agrochemical properties of soils showed that they had a thick humus-accumulative horizon, and were characterized by high organic carbon content (SOC) and neutral acidity. These soils were well supplied with nitrogen and potassium concentrations, but available phosphorus levels were low.

The agrochernozem was characterized by favorable water–physical propertiesand showed good values for water infiltration and moisture categories. The granulometric composition of the soil horizons (Aplow–Bca) was characterized as heavy loamy, while Cca layer was classified as heavy clay. These soils have high potential fertility and are favorable for improving irrigation.

These arable soils did not contain heavy metals or water-soluble salts at concentrations harmful to plants. The availability of micronutrients (lead, cadmium, mercury, cobalt, zinc and copper) was low, whereas manganese content was moderate.

Using the random forest machine learning method, we modelled the spatial distribution and produced digital maps of the soil elements. In general, remote sensing variables were the most effective covariates in predicting soil parameters, which is explained by the spectral reflectance of bare soil. However, relief attributes can also explain the variation of soil properties on heterogeneous terrain and with higher spatial resolution. Thus, significant improvement in the spatial assessment of soil properties can be achieved using unmanned aerial vehicles and a good number of soil samples.

Overall, the assessment and spatial modeling of agrochernozem properties are necessary to monitor the effectiveness of reclamation measures. By tracking changes in soil properties over time, we can determine whether our reclamation efforts are having the desired impact and make adjustments as necessary.

**Author Contributions:** Conceptualization, R.S. (Ruslan Suleymanov) and E.A.; methodology, A.S.; software, A.S.; validation, I.A. and G.G.; formal analysis, I.A.; investigation, R.S. (Ruslan Suleymanov), A.S., G.Z., E.A., I.A. and G.G.; resources, R.S. (Ruslan Suleymanov); data curation, E.A.; writing—original draft preparation, R.S. (Ruslan Suleymanov) and G.Z.; writing—review and editing, R.S. (Ruslan Shagaliev), E.A., A.S. and G.Z.; visualization, G.G.; supervision, R.S. (Ruslan Shagaliev) and E.A.; project administration, R.S. (Ruslan Shagaliev) and E.A.; funding acquisition, R.S. (Ruslan Suleymanov), G.Z. and R.S. (Ruslan Shagaliev). All authors have read and agreed to the published version of the manuscript.

**Funding:** This study was funded by the Ministry of Science and Higher Education of the Russian Federation «PRIORITY 2030» (National Project «Science and University»).

**Institutional Review Board Statement:** Not applicable.

**Informed Consent Statement:** Not applicable.

**Data Availability Statement:** Data are available upon request.

**Conflicts of Interest:** The authors declare no conflict of interest.

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
