# Peer review of "Assessment and Spatial Modelling of Agrochernozem Properties for Reclamation Measurements"

_applsci, doi:10.3390/app13095249_

Round 1

Reviewer 1 Report

This manuscript has been well written in both form and content. I suggest a minor decision with some following comments:

1./ Line 108: Random forest

This section needs to be more precise about the parameters for the RF method such as mtree, ntry, training, and validation ratio data.

2./ I need clarification about Figure 2.

In normal conditions, the water absorption speed negatively correlates to bulk density. But Figure 2, it showed a different way. The bulk density of the soil surface (0-28cm) is less than that of the lower layer, but the water absorption speed of is soil surface is faster than the lower layer. Please explain in more detail in the discussion part.

3./ Line 225-229: I need described data to see how differences of variable importance to each soil feature. 

4./ Table 6: Need the unit of each parameter in the column of RMSE and ME.

Author Response

Dear reviewer, we grateful to you for your careful review of our work.

All corrections were marked up using the “Track Changes” function in the text.

  1. Line 108: Random forest. This section needs to be more precise about the parameters for the RF method such as mtree, ntry, training, and validation ratio data.

Answer: Thanks for the suggestions. We add the information about random forest, including ntree and mtry. The choice of mtry is often the square root of the number of variables. Thus, it was 4 and the parameter ntree was set as 500. We include this information in the text.

Information about cross validation procedure is show in the 2.5. Validation and statistical analyses section. We used A leave-one-out cross-validation (LOOCV) approach, that consists of using all training data leaving one out. That means that our dataset is trained on all the data except for one point and a prediction is made for that point.

  1. I need clarification about Figure 2. In normal conditions, the water absorption speed negatively correlates to bulk density. But Figure 2, it showed a different way. The bulk density of the soil surface (0-28cm) is less than that of the lower layer, but the water absorption speed of is soil surface is faster than the lower layer. Please explain in more detail in the discussion part.

Answer: Thanks for your question. Indeed, the bulk density value in the 0-50 cm layer is less in compare with layer >50 cm. According to the figure, in the upper layer the rate of water absorption is gradual due to heavy loamy texture and high SOC content. The lower horizon (>50 cm) is clayey, compacted and accordingly flows poorly – it’s like the waterproofing. A sharp spike in water absorption in the lower horizon is associated with the fractured structure of the profile and karst. But then, the absorption of water stops due to the clay structure. Therefore, it is necessary to set watering rates. We include this explanation in the discussion part.

  1. Line 225-229: I need described data to see how differences of variable importance to each soil feature. 

Answer: Thanks for the note. We have disclosed this table.

  1. Table 6: Need the unit of each parameter in the column of RMSE and ME.

Answer: Thanks for the note. We added this information.

We thank you again for your review. Please let us know if any parts of the article need to be corrected or improved.

Best regards, team of authors.

Reviewer 2 Report

Comments on the MS: applsci-2343216

Assessment and Spatial Modelling of Agrochernozem Properties for Reclamation Measurements

The MS presents some interesting new findings in the area study, particularly in Table 6 and Fig. 3. However, in my opinion, the MS should be improved as follows.

1.     According to the abstract, ".... The soil quality rating interpretation confirms that these soils have high potential fertility and are convenient for irrigation activities."  Please revise the scientific definition of "convenient" in reflections of the MS results shown in Table 4.

2.     Please include a specific of the new discovery in Table 6 and Figure 3 in the Abstract.

3.    In table 1, please provide wavelength (nm) information for each definition.

4.    Please explain the differences in some of the data(s) between Table 3 and Figure 3. In table 3, for example, K2O was 152-54 mg/kg. However, the prediction in Fig. 3 was around 23.6-12.4 mg/kg, and so on. How intense was the correlation in the data (s)?

5.    The MS should provide a table comparing references of terrace mapping and algorithms, such as in terms of agro-improvement and climate change sustainability. Then, please discuss (suggest) how long the data(s) in Fig. 3 will be changed after some/any land use application, etc.

……………………………………………………………..

Author Response

Dear reviewer, we grateful to you for your careful review of our work.

All corrections were marked up using the “Track Changes” function in the text.

  1. According to the abstract, ".... The soil quality rating interpretation confirms that these soils have high potential fertility and are convenient for irrigation activities." Please revise the scientific definition of "convenient" in reflections of the MS results shown in Table 4.

Answer: Thanks for the suggestion. We added our fundings regarding above-mentioned notes to the abstract.

  1. Please include a specific of the new discovery in Table 6 and Figure 3 in the Abstract.

Answer: Thanks for the question. We improved our abstract part and disclose our results.

  1. In table 1, please provide wavelength (nm) information for each definition.

Answer: Thanks for the comment. We added this information for Sentinel-2A bands.

  1. Please explain the differences in some of the data(s) between Table 3 and Figure 3. In table 3, for example, K2O was 152-54 mg/kg. However, the prediction in Fig. 3 was around 23.6-12.4 mg/kg, and so on. How intense was the correlation in the data (s)?

Answer: Many thanks for the valuable note. Its mistake relates with units (mg/100 g soil instead mg/kg) in the legend map. We fixed it for N, P and K content.

Also, we conducted digital mapping using 40 soil samples from 0-10 cm layer. Therefore, the numbers are little bit other than in the 0-28 cm horizon.

  1. The MS should provide a table comparing references of terrace mapping and algorithms, such as in terms of agro-improvement and climate change sustainability. Then, please discuss (suggest) how long the data(s) in Fig. 3 will be changed after some/any land use application, etc.

Answer: Thanks for your suggestions. Indeed, digital mapping of soil properties of agricultural lands differs from mapping of large areas (regional and global scale). In general, adequate comparing different methods for digital mapping is a difficult task, since it depends on different methods as well as on soil conditions, types of tillage, fertilization, etc. There is a review article (https://doi.org/10.1088/1748-9326/aca41e) where its discussed.

Therefore, we focused on comparing the application of digital mapping methods specifically on arable chernozems (Suleymanov et al 2021; Sahabiev et al 2021). So, to create a reliable comparison with other studies its difficult task and may by unbiased. Also, we added the comparison of other studies related with digital mapping of soil parameters.

Understanding changes in soil properties and their variation due to irrigation is an important task. We added the paragraph in the discussion about changes of soil properties and its variability due to irrigation and other agriculture practices.

We thank you again for your review. Please let us know if any parts of the article need to be corrected or improved.

Best regards, team of authors.

Reviewer 3 Report

Comments to Author

I have reviewed the manuscript entitled ‘Assessment and Spatial Modelling of Agrochernozem Proper ties for Reclamation Measurements’. This is a time demanding manuscript. They tried to find out the soil properties by digital mapping using remote sensing. I appreciate to publish this kind of manuscript.

However, I have some observation to publish this manuscript.

Ø  Introduction is poor. You have to improve introduction

Ø  You have to clarify why you select this area?

Ø  Already this type of work has been done. So what is the specialty of your research? You have to mention in Introduction.

Ø  Use note below the table

Ø  Fig. 2: time 0-100 min is okay. However 100-400 min. Have any significant difference. If possible show statistically difference.

Ø  Fig.3 Pic are not clear. If possible use more clear photos.

Ø  Discussion should be chronologically present. If possible use more recent references

Ø  Author should avoid the long sentences. Make the long sentences to simple sentences.

Ø  Conclusion should be re-write. Present precisely.

Ø  Serious problems related to references. Most of the references are old. You have to use current references.

Ø  Some of the references are irrelevant. Use relevant references.

Ø  Authors should read the authors guideline carefully. They make some minor mistake due overlook the author guideline.

Ø  Follow the Author guidelines for ‘References’.

Ø  Also see the attached manuscript for minor corrections.

Moderate editing of English language is necessary.  

Author Response

Dear reviewer, we grateful to you for your careful review of our work.

All corrections were marked up using the “Track Changes” function in the text. In the reference part we marked corrections by yellow color.

  1. Introduction is poor. You have to improve introduction

Answer: Thanks for the suggestion. We improved the introduction part.

  1. You have to clarify why you select this area?

Answer: Thanks for the question. We selected this study plot for several reasons:

Due to climate change, the republic involves arable land in irrigation. Especially suitable for this are plots with flat terrain and located near the river. So, the study of this particular site was agreed to prepare it for irrigation. We added this explanation in the text.

  1. Already this type of work has been done. So what is the specialty of your research? You have to mention in Introduction.

Answer: Thanks for the suggestion. We added this explanation this in more detail in the introduction.

  1. Use note below the table

Answer: Thanks for the comments. We fixed this part.

  1. 2: time 0-100 min is okay. However 100-400 min. Have any significant difference. If possible show statistically difference.

Answer: Thanks for the comment. We performed this analysis in two replicates, so the statistical difference would not be clear. In general, its difference related with clayey and compacted lower horizons, as well as absence of structural aggregates. Therefore, the speed of absorption is different between top and >50 cm layer. We also added this information in the discussion part.

  1. 3 Pic are not clear. If possible use more clear photos.

Answer: Figure 3 shows the generated digital maps made by the RF method within the boundaries of the site. For a better understanding of the spatial distribution of properties, you can also look at Figure 1. We're having a little trouble improving it. Please let us know if you can be more specific about improving it.

  1. Discussion should be chronologically present. If possible use more recent references

Answer: Thanks for the comment. We have set the order of the discussion according to the results part: morphological description of the soil profile, chemical, physical properties, heavy metals and digital mapping. Please let us know which part we should replace. We added the more recent references in the manuscript.

  1. Author should avoid the long sentences. Make the long sentences to simple sentences.

Answer: Thanks for the comment. We checked the long sentences and rewrote them.

  1. Conclusion should be re-write. Present precisely.

Answer: Thanks for the comment. We rewrote the conclusion and highlighted the soil assessment results.

  1. Serious problems related to references. Most of the references are old. You have to use current references.

Answer: Thanks, we added the more recent references in the manuscript. Also, we eliminated the sources Sokolov et al. 1975 and Orlov et al. 1985.

At the same time, we would to remain the other references, as the article (Gabbasova et al 2006) is closely related with our study for following reasons. Firstly, this work related with the same soil type (it’s the same region) and is aimed at irrigation assessment. So, its valuable reference that allows us to compare our fundings. As irrigation continues to be used on these soils, we do not consider these results as outdated.

The second article by Breiman (2021) is fundamental to the random forest method with over 100,000 citations. We also would to remain this work.

  1. Some of the references are irrelevant. Use relevant references.

Answer: Thanks. We also added the relevant literature in discussion part.

  1. Authors should read the authors guideline carefully. They make some minor mistake due overlook the author guideline.

Answer: Thanks for the suggestion. We checked the article and fixed some mistakes.

  1. Follow the Author guidelines for ‘References’.

Answer: Thanks. We fixed the references.

  1. Also see the attached manuscript for minor corrections.

Answer: Thanks for the provide file. We check the attached file and fixed corrections. However, according to the Author guidelines for ‘References’ – In the text, reference numbers should be placed in square brackets [ ] and placed before the punctuation; for example [1], [1–3] or [1,3]. Also, bold is used for years of articles, not books or other sources. So, we correct the years for articles.

We thank you again for your review. Please let us know if any parts of the article need to be corrected or improved.

Best regards, team of authors.